# Evaluation of Thermochemical Treatments for Rice Husk Ash Valorisation as a Source of Silica in Preparing Geopolymers

**DOI:** 10.3390/ma16134667

**Published:** 2023-06-28

**Authors:** Noelia Bouzón, Alba Font, María Victoria Borrachero, Lourdes Soriano, José Monzó, Mauro M. Tashima, Jordi Payá

**Affiliations:** Institute of Concrete Science and Technology (ICITECH), Universitat Politècnica de València, 46022 València, Spain; noelia_bouzon@hotmail.com (N.B.); fonpeal@gmail.com (A.F.); vborrachero@cst.upv.es (M.V.B.); lousomar@upvnet.upv.es (L.S.); jmmonzo@cst.upv.es (J.M.); maumitta@upvnet.upv.es (M.M.T.)

**Keywords:** rice husk ash, sodium silicate, filtering/gravimetric method, mechanical strength, silica dissolution

## Abstract

The use of geopolymers has revolutionized research in the field of construction. Although their carbon footprint is often lower than that of traditional mortars with Portland cement, activators such as sodium silicate have a high environmental impact in the manufacturing of materials. Employing alternative alkali sources to produce geopolymers is necessary to obtain materials with a lower carbon footprint. The present research explores the use of rice husk ash (RHA) as an alternative source of silica to produce alkaline activators by four methods: reflux; high pressure and temperature reaction; thermal bath at 65 °C; and shaking at room temperature. To evaluate the efficiency of these methods, two types of experiments were performed: (a) analysing silica dissolved by the filtering/gravimetric method; and (b) manufacturing mortars to compare the effectiveness of the treatment in mechanical strength terms. The percentages of dissolved silica measured by the gravimetric method gave silica dissolution values of 70–80%. The mortars with the best mechanical strength results were the mixtures prepared with the thermal bath treatment at 65 °C. Mortar cured for 1 day (at 65 °C), prepared with this activator, yielded 45 MPa versus the mortar with commercial reagents (40.1 MPa). It was generally concluded that utilising original or milled RHA in preparing activators has minimal influence on either the percentage of dissolved silica or the mechanical strength development of the mortars with this alternative activator.

## 1. Introduction

The construction industry is responsible for about 8% of global anthropogenic CO_2_ emissions, and it is the mainly the manufacturing of Portland cement that produces the most emissions. Some authors estimate that around 842 kg of CO_2_ is produced for every ton of generated clinker. Of these emissions, 60% are due to limestone calcination [1]. To meet the cement industry’s 2050 carbon neutrality objective, different processes must be developed. Some advocate using cements with lower clinker content, such as LC3 cement [2,3]. Another of the most widely studied approaches in recent decades has been to implement so-called alkaline-activated materials or geopolymers [4,5,6].

When manufacturing geopolymeric systems, most environmental and cost production problems are attributed to the alkaline reagent, more specifically to the production of sodium silicate (Na_2_SiO_3_) and NaOH to a lesser extent. The environmental impact of Na_2_SiO_3_ production is the most important parameter in calculating global warming [7]. As for the cost of producing geopolymers that use commercial activators, their production price can be higher compared with the production of traditional materials with cement [8]. Consequently, employing alkaline activating reagents derived from waste can be an interesting solution to reduce these environmental and economic problems.

Some silica-rich residues like those from glass have been used to prepare alkaline activators that substitute Na_2_SiO_3_ [9,10]. For example, Puertas and Torres-Carrasco [9] worked with glass waste with a percentage of SiO_2_ at around 70%. They mixed the residue with NaOH/Na_2_CO_3_ for 6 h at 80 °C. Then this mixture was filtered, and the solution was used as an activating solution. These authors activated pastes with blast furnace slag (BFS) and compared the obtained results to the alternative solution, and also to the results obtained with either Na_2_SiO_3_ or NaOH/Na_2_CO_3_. The paste with the higher compressive strength (83 MPa) was obtained with Na_2_SiO_3_, compared with 66 MPa and 44 MPa obtained for the pastes activated with the solution from glass waste and NaOH/Na_2_CO_3_, respectively.

García Lodeiro et al. [8] studied the replacement of NaOH using a wasted cleaning solution from the aluminium industry. As a precursor, they employed BFS. The pastes activated with a 4 M NaOH solution yielded 29.9 MPa after 28 curing days versus 33.4 MPa obtained by the paste activated with the waste cleaning solution. 

Employing agricultural waste as biomass is a good way to reduce the amount of oil and coal used to produce energy. Biomass ash production may cause serious secondary pollution problems [11]. Biomass ashes have been studied in construction materials in recent decades. For example, rice husk ash (RHA) or sugar cane bagasse ash (SCBA) have been investigated as pozzolans in Portland cement concrete and mortars. Moraes et al. [12] used SCBA at percentages up to 30% to substitute cement. They obtained a similar compressive strength to the mortar with only Portland cement. The high quantity of silica content in RHA promotes enhanced durability and mechanical properties [13,14,15]. 

There are some new uses of biomass ashes in construction materials. Biomass ashes may be utilised to prepare alkaline solutions when fabricating geopolymers. Ashes rich in SiO_2_ are used to prepare alternative sodium silicate [16,17,18,19,20] and ashes rich in K_2_O can act as a substitute of KOH [21,22,23,24].

Rice was the third most produced agricultural crop in 2019, and 756 million tons were produced globally [25]. Approximately 0.20 ton of ash is generated per ton of calcined husk. As rice cultivation occurs in many places on our planet [14], utilising RHA would be applicable in many countries.

Its use as a source of silica as an alternative to sodium silicate has been explored by several research groups. The first reference about RHA use to fabricate alternative silicate was published by Bejarano et al. [26]. These authors proposed a hydrothermal process between NaOH and RHA at 100 °C for 2 h. They showed that both amorphous silica and much of the crystalline fraction present in RHA had dissolved at this temperature. Subsequently, the researchers of this group applied this method to systems with metakaolin (MK) as the precursor and KOH/RHA as the activator [27], or with BFS and FA, (12) and obtained very positive results in both cases. 

Tchakouté et al. [28] prepared alternative sodium silicate by mixing RHA and NaOH at different SiO_2_/Na_2_O molar ratios. The mixture was mixed with water and placed inside a magnetic stirrer for 2 h at 80 °C. Then the suspension was filtered and the obtained liquid was stored at ambient temperature for at least 1 day before the solution was used. They employed MK as a precursor and the prepared pastes were cured for 28 days at room temperature. The higher compressive strength value (36.29 MPa) was obtained for the paste activated at the 1.25 SiO_2_/Na_2_O molar ratio versus the paste at the 0.31 ratio, which only yielded 3.58 MPa.

A different methodology from those set out above is that proposed by Rajan and Kathirvel [29]. These authors ground sodium hydroxide flakes to obtain a powder and then mixed it with RHA for 5 min. This mixture was heated in an oven at 100 °C for different time intervals (1, 2 and 3 h) and variations of the NaOH/RHA mass ratio (0.5/1.0, 1.0/1.0, and 1.5/1.0). The alternative silicate was employed as an activator in the mixtures with BFS as precursor at the ratio of 3 parts activator to 10 parts BFS. They concluded that the optimum synthesis time was 2 h for all the NaOH/RHA ratios. The optimum NaOH/RHA ratio was 1.0/1.0. With these two optimum parameters, a system with 60 MPa compressive strength was obtained after curing for 28 days.

In the present paper, the objective was to explore the use of several methods to dissolve silica from RHA by reacting with NaOH to prepare an alternative sodium silicate reagent. The thermochemical treatments carried out for dissolving silica in RHA were: reflux (R); high pressure and temperature reaction (HPT); thermal bath at 65 °C (65C); and shaking at room temperature (RT). To verify the effectiveness of the proposed methods, a gravimetric study was carried out to quantify the dissolved percentage of RHA, followed by a check to see if the higher percentage of dissolved RHA corresponded to higher mechanical strength in the mortars manufactured with the alternative silicate using catalytic cracking catalyst (FCC) waste as a precursor. The suspensions that resulted after the different thermochemical treatments were used without filtration. As a reference, a mortar made with a mixture of commercial Na_2_SiO_3_ and NaOH was employed as activator. Although RHA has been previously reported by several authors as a source of silica combined with NaOH for the activation of different precursors, the novel aspect of this study is the development and comparison of four thermochemical treatments for silica dissolution from RHA that take into account the following variables: the particle size of RHA, the SiO_2_/Na_2_O and H_2_O/Na_2_O molar ratios when preparing the activator, temperature, and the reaction time. Thermogravimetry analysis and field emission scanning electron microscopy (FESEM) were applied to assess some of the results. 

## 2. Materials and Methods

X-ray fluorescence (XRF), particle size distribution (PSD), powder X-ray diffraction (XRD) and FESEM were used to characterise the starting materials. XRF was performed using Philips Magic Pro—PW2400. Mastersizer 2000 equipment (Malvern Instruments S.L., Malvern, UK) was employed to obtain the PSD and granulometric parameters (d_mean_, d_(0.1)_, d_(0.5)_ and d_(0.9)_) with samples suspended in deionised water for taking measurements. The XRD evaluation was performed with Brucker AXS D8 Advance, which generates the X-ray (20 mA and 40 kV). XRD patterns were obtained between 5° and 70° of 2θ, with a 0.02° angle step in a 2 s cumulative time. FESEM observation was performed under a ZEISS model ULTRA 55 electronic microscope (Jena, Germany) and the samples were covered with carbon.

Thermogravimetry equipment was used to characterise some of the gels obtained when preparing the RHA-based activators. A Mettler-Toledo TGA 850 device was employed for the thermogravimetric analysis, which was equipped with a microbalance (0.1 μg). The sample was placed inside a 70 μm capacity alumina crucible and covered by a lid with a hole. The selected test parameters were: temperature range of 35–300 °C, 20 °C/min heating rate, and dry air atmosphere at a 75 mL/min gas flow rate.

Commercial reagents (NaOH and Na_2_SiO_3_) were obtained to prepare the reference activator. NaOH (98% purity) was provided by Panreac S.A as pellets. The commercial sodium silicate was composed of 8% Na_2_O, 28% SiO_2_ and 64% H_2_O (% by mass) at a pH of 11–11.5. The company that supplied Na_2_SiO_3_ was Merck (Spain).

FCC was employed as a precursor and was provided by OMYA Clariana S.A. (Tarragona, Spain). This residue comes from the petrochemical industry and is mainly composed of silica (SiO_2_ = 47.76, wt%) and alumina (Al_2_O_3_ = 49.26, wt%) (from the XRF analysis; see Table 1). FCC was supplied as ground material and its mean particle diameter (d_mean_) was 17.12 µm.

RHA was provided by DACSA S.A (Tavernes Blanques, Spain). This company has a power cogeneration system where rice husk is employed as fuel to generate alternative green energy and RHA is a residue. As Table 1 shows, silica constitutes 85.58% of the RHA, which confirms the viability of silica source valorisation in preparing an alternative alkali activator. In rice farming, the amount of K_2_O depends on the type and proportion of the employed fertilisers [30]. The studied RHA had 3.39% potassium oxide, which is a high percentage compared with the other minor compounds. The obtained loss of ignition (LOI, Table 1) value was 6.99%. LOI was due to the presence of unburned carbonaceous particles. The high combustion temperature of the rice husk (> 1000 °C) and the burning rate with the presence of K_2_O caused RHA melting to partially trap unburned carbon [31].

To determine the amorphous and crystalline percentages of RHA, a method that combined acid and basic attack was followed. The methodology of this process is described by Hidalgo et al. [32] and utilises rice straw ash (RSA) as the pozzolanic material. The obtained values showed that RHA had 31.5% amorphous silica and 49.7% crystalline silica. The sum of both silica percentages was slightly lower than the quantity of silica obtained by XRF (see Table 1). This was attributed to the several acid/base treatment steps in the chemical analysis, where the error was larger. 

In order to assess the influence of RHA granulometry on the preparation of the RHA-based activator (the amount of dissolved silica and its reactivity), and in the geopolymer mechanical properties, ash was employed in two forms: (i) the original RHA (RHA-O); (ii) ash milled for 4 h (RHA-M). The milling procedure was carried out in an industrial mill. The particle size distribution analysis for RHA-O found that most particles were between 10 μm and 200 μm (Figure 1a). For RHA-M, the particle size was smaller, with particle diameters of 1–100 μm (Figure 1a). The d_mean_ values were 62.24 μm and 20.31 μm for RHA-O and RHA-M, respectively. 

Figure 1b represents the X-ray diffractogram for RHA-M. It depicts the presence of peaks corresponding to some crystalline products, such as tridymite, cristobalite, quartz and sylvite. Additionally, a baseline deviation between 15° and 30° of 2θ appears, which indicates the presence of amorphous material.

The FESEM micrographs of RHA-O and RHA-M are shown in Figure 2. RHA-O (Figure 2a,b) presented the typical morphology of this ash, as previously reported by the authors [26,30]: irregular shape with porous and irregular structure with the presence of a rigid “skeleton” of silica, which was not destroyed during combustion. After the milling treatment, RHA-M (Figure 2c,d) showed notably smaller particles with an irregular shape and low porosity, in which the original silica-skeleton structures had been destroyed.

Both the above-mentioned RHA forms, original (RHA-O) and milled (RHA-M), were employed for preparing the activator as an alkaline aqueous suspension. 

Two types of alkali suspensions were prepared for each proposed thermochemical treatment: (a) a smaller volume sample (3 g of NaOH, 2.9 g of RHA, 10 mL of deionised water) used to calculate total dissolved ash; and (b) a bigger volume sample (81 g NaOH, 78.8 g of RHA, 270 g of tap water) employed to prepare geopolymeric mortars. 

The time spent running each treatment was selected by taking into account two aspects. The first was the temperature of the medium in which RHA was dissolved: for high-temperature methods (R and HPT), the time was shorter than 500 min; for moderate-temperature methods (65C), it was less than 700 h; the mildest method (RT) took <225 days. Secondly, data were collected until a decrease in dissolved RHA or a constant percentage was observed. Each thermochemical treatment was carried out according to the following descriptions:Reflux (R)

The prepared NaOH (7.5M) solution was mixed at room temperature with the solid RHA inside a round bottom flask connected to a reflux condenser to keep the suspension volume constant. A 50 mL flask was employed to determine the RHA dissolution percentage and a 500 mL flask was used for preparing the activator to be used during mortar manufacturing. The flask with the NaOH/RHA mixture was placed on a heating mantle and stirred with a magnetic stirrer. The stirrer bar allowed the constant homogenisation of the mix during treatment. The mix was preheated to reach the boiling point (around 110 °C due to the high NaOH concentration) and then the reflux period began. The refluxing times were: (i) 5, 10, 15, 20, 30, 60, 90, 120, 180, 240 and 480 min to study the percentage of dissolved RHA; and (ii) 15, 30, 60, 90, 120, 180, 240 and 480 min for the FCC activated mortars. This treatment was named “R-Xz”, where X is the reflux time and z is the unit time (z = (m) minutes).

High pressure and temperature (HPT)

RHA was mixed with the H_2_O and NaOH pellets (the temperature reached 90 °C due to the dissolution heat of NaOH) inside a glass container with a screw cap. The container was placed in an oven at 110 °C. The container was not subjected to shaking. The experimental times were: 15, 30, 60, 90, 120, 180, 240 and 480 min. For both studies (dissolved RHA and manufactured geopolymer mortars), the same times were assessed. This treatment was called “HPT-Xz”, where X is the treatment time and z is the unit (z = (m) minutes).

Thermal bath at 65 °C (65C)

For this treatment, RHA was mixed with the NaOH solution (7.5M) at 65 °C inside a polyethylene bottle. The bottle was left in a thermal bath at 65 °C for the established treatment times: 1, 3, 6, 12, 24, 60, 168, 336, 504 and 672 h (for both studies: dissolved RHA and mortar manufacturing). The polyethylene bottle was manually shaken for 1 min every 24 h. This treatment was called “65C-Xz”, where X is the treatment time and z is the unit (z = (h) hours).

Room temperature (RT)

When the NaOH/H_2_O solution reached room temperature after dissolution of the pellets, it was mixed with RHA and poured inside a polyethylene bottle. The bottles were manually shaken for 1 min every day on the first 28 days and every 3–4 days after this stage. 

The selected study times of the dissolved RHA were 1, 3, 7, 10, 14, 21, 28, 35, 42, 56, 63, 98, 112, 140 and 224 days. The treatment times of the RT treatment for mortar manufacturing were 1, 3, 7, 14, 21 and 28 days. This treatment was named “RT-Xz”, where X is the treatment time and z is the unit (z = (d) days).

Table 2 summarises the treatment identifications and the times selected for each to study not only the dissolved RHA, but also the effect of the alternative activator on the flexural and compressive strengths of the geopolymeric mortars.

With the smaller volume suspensions, a filtering/gravimetric assessment was carried out to calculate the percentage of dissolved RHA after each thermochemical treatment [17]. The resulting suspension was filtered by a vacuum system and the retained solid was washed with deionised water. The solid residue was dried at 60 °C until constant mass. The difference between the employed ash (2.9 g) and this weight can be considered the silica dissolved with treatment. It must be pointed out that other RHA compounds could have dissolved. Ziegler et al. [33] measured the percentage of calcium, sodium, phosphorous and silicon dissolved from an RHA sample using 8M KOH solution. They reported the effect dissolution after 4, 24 and 168 h in alkaline medium at room temperature. After 168 h, Ca and Na showed very limited solubility, with only ca. 1% dissolved, and P dissolved by about 26%. However, ca. 91.3% of the silicon dissolved. 

As the SiO_2_/Na_2_O molar ratio of the activator is one of the most important parameters in the geopolymerisation process [34,35] the influence on the dissolved silica with varied RHA and NaOH ratios was studied. For this study, the selected thermochemical treatment was R-60m. SiO_2_/Na_2_O ratio was selected by: (a) varying the amount of RHA in the mixture while maintaining the amount of NaOH in the suspension constant; and (b) decreasing the amount of NaOH while maintaining RHA. The SiO_2_/Na_2_O molar ratios for both approaches are listed in Table 3.

The mortars were composed of FCC, the activator and natural sand. After the thermochemical treatment, no filtration of the RHA-based activator was performed. As the control mortar, a traditional alkali solution (Na_2_SiO_3_/NaOH/H_2_O) was employed for the FCC activation at the same dose as the alternative RHA-based activator (SiO_2_/Na_2_O = 1.17, H_2_O/Na_2_O = 14.81 and H_2_O/FCC = 0.6). The FCC/sand mass ratio was 1/3. Table 4 shows the dosage of control mortar and RHA mortar. (The quantity of materials of the mortars with RHA is the same for all treatments.) The mechanical strength development of the control mortar was obtained for 65 °C curing temperature and for room temperature: these values were 55.0 MPa (28 days) and 48.2 MPa (7 days), respectively. These data demonstrated the stability of the alkali-activated FCC.

The mortar was mixed for a total time of 4.5 min. Then the mixture was poured inside a 4 × 4 × 16 cm^3^ prismatic mould and vibrated for 2 min. The moulds were covered with plastic film to avoid water evaporation and carbonation and were then placed inside the thermal bath (65 ± 2 °C). The samples were demoulded after 4 h in the thermal bath. Then the mortar specimens were returned to the curing place until the testing age (24 h).

The new alternative geopolymer mortars (with the use of RHA) were labelled by the thermochemical treatment according to Table 2, by which the alternative activator was manufactured. The control mortar, prepared with commercial NaOH and Na_2_SiO_3_, was labelled as “C”. The dosages for the new mortars produced for the study of the SiO_2_/Na_2_O molar ratio are shown in Table 5. The quantities of sand, FCC and water are the same as for the mortars shown in Table 4. 

The mortars were mechanically tested and compared. Flexural strength (Rf) was obtained by testing three prismatic samples. Then the compressive strength (Rc) of the six resulting portions was measured (according to Standard UNE 196-1 [36]).

## 3. Results and Discussion

### 3.1. Study of the Dissolved RHA

Figure 3 displays the results of the dissolved RHA (dissolved silica) obtained with each treatment. In all cases, the percentage of dissolved RHA tended to be higher than 70%. This value was significantly higher than the percentage of amorphous silica present in RHA (31.5%), which means that during the thermochemical processes, part of the crystalline silica also dissolved.

During R, when RHA-O as RHA-M were assessed, the dissolved silica sharply incremented for the reflux time < 60 min. From R-5m to R-60m, the dissolved silica progressively increased from 50% to 80% for both RHA types. 

After 60 min, the dissolved silica percentage remained within the same range of values up to 120 min for both RHA-O and RHA-M. For RHA-O, the longest refluxing times (R-180m, R-240m and R-480m) resulted percentages of dissolved silica within the 70–80% range. During RHA-M, the decreases in the dissolved silica were marked, ranging from 77% (R-120m) to 35% (R-240m). Finally, the value increased to 61% for R-480m.

RHA-M behaviour in the refluxed treatment, which ranged between 120 and 240 min, was attributed to the ash fineness and to a gelation process. In Figure 4, the FESEM micrographs (both at the same magnification) of the retained solid residue obtained in R-240m with RHA-O and RHA-M are compared. Significant differences are observed: For RHA-O, there is a particulate structure. For RHA-M, a gel-type structure of the residue was confirmed. The dissolved silica was gravimetrically obtained, and the gel could not be filtered but was retained in the filter, which resulted in decreased dissolved silica in the assessment.

These results were confirmed by the thermogravimetric analysis of the R-240m residues with both RHA. The DTG curves were compared (Figure 5). Between 100 °C and 200 °C, a peak appeared for both the R-240m residues. This is related to mass loss due to adsorbed water in the solid, with a greater mass loss for RHA-M (7.1%, peak at 156 °C) than for RHA-O (4.0%, peak at 168 °C). The DTG curve for the unprocessed RHA-O does not show any mass loss in this range. The FESEM and thermogravimetric analysis characterisation results confirmed the gelation process, which suggests that silica was dissolved, mainly for RHA-M. However, part of it evolved to gel, which was retained in the filter paper.

The evolution of the dissolved silica that resulted in HPT was similar to that observed in R for both RHA types (Figure 3b). For the shortest treatment time (15 min), the dissolved silica was only 30–35%, which was significantly lower than for R (50–55% for the same time). This was attributed to not shaking the mixture in HPT. For the first 90 min of treatment, the percentage of dissolved silica increased from 30% to values close to 80%. In the original RHA, this percentage remained constant (fluctuating between 70 and 80%) until 480 min of treatment. For RHA-M, a decrease in the dissolved silica occurred from HPT-90m (75%) to HTP-240m (41%), followed by a new rise in HTP-480m (68%). In this treatment, the gelation effect on RHA-M had an influence for times longer than 90 min. Apparently, the formed gel re-dissolved under a longer treatment time (480 min) and in the same way as for R.

Figure 3c shows the results for 65C. During the first 24 h of treatment, the dissolved silica progressively increased to values of around 80% for both RHA types. The results obtained from 24 h to 28 days fell within the range of 60% to 80% of dissolved silica for both RHA-O and RHA-M, but the latter showed wider fluctuations. Despite the milder conditions in 65C compared with R and HPT, a high dissolved silica percentage (80%) was obtained after a relatively short period. 65C was less energy-intensive, which suggests that 65C is a very interesting thermochemical procedure in economic terms.

When RT was applied (Figure 3d), the dissolved silica increased for the first 14 days. At this reaction time, 59% dissolved silica was reached for RHA-O and 74% for RHA-M. Therefore, for this thermochemical treatment, ash fineness is crucially important in reaction rate terms. For RT, the gelation process was insignificant and the percentage of dissolved silica fluctuated during the 14–224-day period. This was the treatment with the lowest power consumption, but a longer period (minimum of 14 days) was needed to reach 70% dissolved silica. Under RT, the amount of dissolved silica showed a slower increase in time, with values (close to 70%) that were slightly lower than those obtained by the more energy-intensive methods (R, HPT and 65C). 

Finally, the influence of the relative proportion of RHA and NaOH on the dissolved silica was studied for R-60m and the results appear in Figure 6.

The increased silica content in the alkaline activator promotes a reduction in the geopolymerization reaction rate. This fact was also reported in previous investigations where systems with high silica content yielded an early paste solidification before a complete geopolymerization reaction [37]. On the other hand, the sodium concentration in the alkaline activator should be enough to allow the charges to balance for the siliceous tetrahedra’s substitutions by aluminium, but to avoid sodium carbonate formation, it should not be in excess [38,39].

When RHA-O was employed at the lower SiO_2_/Na_2_O ratio (decreased RHA dose), silica dissolution was 93.10%, which implies total silica dissolution taking into account the LOI for this ash (6.99%; see Table 1). This behaviour can be explained by the decrease in the relative amount of silica and as the NaOH concentration was constant, the dissolution process was more effective for this reaction time. With a SiO_2_/Na_2_O ratio higher than 1.17, when the amount RHA was increased in the mix, the dissolved silica remained within the same range, at around 80%. 

When RHA-M was employed, the amount of dissolved silica came close to 80% for the SiO_2_/Na_2_O ratio that equalled or was lower than 1.17. However, with a larger amount of RHA-M but with the same NaOH, the percentage of dissolved silica dropped to 55–60%. Once again, such behaviour was due to the gelation process, which was more marked with ash fineness.

When the variation in the SiO_2_/Na_2_O ratio was due to the reduction in NaOH, the percentages of dissolved silica remained within the 70–80% range when both RHA-O and RHA-M were employed. These results highlight an opportunity to manufacture an alternative activator with silica from RHA and to lower the NaOH concentration to 50%. This reduction in the commercial chemical reagent could involve important economic and environmental savings when developing new binding materials.

Regarding the dissolved silica values obtained in this research work and those obtained by other authors ([16,26,40], several aspects must be examined. Bouzón et al. [17] et al. demonstrated that quartz (material with 100% crystallinity) was attacked in an alkaline medium at high temperature, and its dissolved silica values were around 40%. Bejarano et al. [26] manufactured an alternative sodium silicate using two samples with different SiO_2_ crystalline contents at 100 °C for 2 h. The RHA with the smaller amount of amorphous phase (27.7%) obtained 90% dissolved silica. These results corroborate those herein reported, and demonstrate that both crystalline and amorphous silica may be dissolved under suitable conditions. Tong et al. [40] applied hydrothermal treatment to obtain silicate from RHA using different NaOH molarity, temperatures and time values. These authors obtained percentages of dissolved silica between 85 and 95%, but they did not present the proportions of the amorphous and crystalline phases.

### 3.2. Studying an Alternative Alkaline Solution in Geopolymeric Mortars

In this section, we discuss the mechanical mortar strengths (flexural Rf and compressive Rc) obtained when applying the activator produced by the different treatments analysed in the previous section. For this purpose, a control mortar, based on FCC as a precursor and prepared with commercial reagents (sodium silicate and NaOH), was used for comparison purposes. All the mortars were cured for 24 h at 65 °C. For the mortars prepared with an alternative RHA-based activator, the mixtures obtained from the thermochemical treatments were not filtered and the suspensions were used as obtained (they were left at room temperature whenever necessary).

Figure 7 shows the mechanical development of the FCC-based geopolymeric mortars with alternative activators prepared by means of the R thermochemical treatment. In general, these mortars displayed similar mechanical behaviour (Rc and Rf) for all the refluxing times and both RHA samples. These results were obtained when both RHA-O and RHA-M were employed. Apparently, the gelation process did not affect the mechanical development of mortars, probably because the gel also reacted to the precursor. When comparing the mechanical results to the control geopolymer, the flexural strength for the mortar with an alternative activator yielded higher values than the control. The compressive strength for the mortars with the RHA-based activator was slightly lower than the control mortar, except for the R-15m treatment, for which the difference in R-240m for RHA-M was slightly larger than for the control. Probably, the presence of small undissolved particles in the treated RHA enhanced the microstructure of the cementing matrix and decreased the cracks or the continuity of the pore network in the sample [41]. This effect was not present in samples activated with commercial sodium silicate solution. These compressive and flexural strength results demonstrate that treatment R is a feasible option to prepare activators with less commercial chemical reagent consumption.

The obtained mechanical behaviour for the mortars prepared with the HPT activator was different from treatment R (Figure 8). In general, the mechanical values were lower for the activators prepared with the 15 and 30 min reaction times. This agrees with the low dissolved silica percentage shown in Figure 3b. For longer treatment times, similar results to those obtained in the treatment R were observed. This mechanical behaviour suggests that shaking the activator during the thermochemical reaction was significantly important.

With the mortars based on the activator from 65C (Figure 9), a progressive increase in Rf up to 24 h occurred, and then similar values were obtained at 672 h. For the control sample, the Rf values obtained after 6 h of treatment were higher with both RHA-O and RHA-M. The Rc evolution was similar to that described for Rf during the first 24 h, but Rc evolution continued progressively until 672 h. This evolution was better when RHA-O was employed. For this thermochemical treatment, such behaviour means that RHA-O performed better and milling was unnecessary for preparing a reactive activator. Additionally, the best compressive strength results were obtained for these activators, especially for RHA-O. In Figure 3d, samples with RHA-O showed higher dissolved silica, especially for treatment times longer than 24 h, and this fact could yield better strength development.

The mechanical behaviour results after 24 curing hours at 65 °C of the mortars based on an alternative alkaline dissolution manufactured by RT are reported in Figure 10. In this case, the Rf and Rc values were generally higher for the samples with RHA-M than for those with RHA-O under treatment times lasting 7–28 days. These results agree with the behaviour noted in the dissolved silica calculations with RT (Figure 3d). In general, it can be stated that for this thermochemical treatment with the lowest energy consumption, the process requires a long time and previous RHA milling, which is critical for the dissolution rate and also for strength development.

Regarding the study on the influence of the relative amounts of RHA and NaOH (see Table 3), the mechanical results are shown in Figure 11 and Figure 12, respectively.

For the series in which the amount of RHA varied, the proportion of silica (Figure 6a) was slightly higher for a low SiO_2_/Na_2_O molar ratio (e.g., dissolved 0.58). However, the absolute amount of silica in the activator was small. Consequently, the flexural and compressive strengths (Figure 11) were lower (below 10 MPa). However, when the RHA dose increased (SiO_2_/Na_2_O molar ratio = 1.17 or 1.39), the dissolved silica remained constant for RHA-O and slightly decreased for RHA-M. (As previously explained, the gelation effect influenced the dissolved silica calculation.) In this case, the absolute amount of dissolved silica in the activator increased and, consequently, the mechanical behaviour improved. Tchakouté et al. [28] obtained similar results by applying the highest mechanical strength for the mixture with the highest SiO_2_/Na_2_O ratio, which was 1.25 in their case. When comparing both ashes, RHA-M yielded better results despite the gelation process, as shown by the values corresponding to the SiO_2_/Na_2_O molar ratio = 1.74. For this precursor and mortar design, apparently there is an optimum SiO_2_/Na_2_O molar ratio value. Similar results were presented by Tashima et al. [34]. The mechanical strength development of FCC activated with the R-60m activator was checked in order to confirm the stability of the cementing products with curing time: the mechanical strength of mortar cured after 7 days at 65 °C was 44.9 MPa, and after 28 days at room temperature it was 35.9 MPa. The value for 7 day/65 °C (44.9 MPa) increased about 20% with respect to that obtained after 1d/65 °C (ca. 35 MPa, see Figure 11a). This indicates that under a longer curing time, the reaction progressed and more hydrates were formed. This behaviour confirms that the use of the alternative activator based on RHA/NaOH yields an excellent development of the cementing matrix after long curing times.

When the NaOH concentration percentage was lowered when preparing the activator (See Figure 6b), the proportion of dissolved silica was similar. Nevertheless, the strength development was worse (see Figure 12) in both the flexural and compressive tests. Rf and Rc progressively decreased for the RHA-O- and RHA-M-based activators. The obtained values with 50% less NaOH (SiO_2_/Na_2_O molar ratio = 2.30) were noticeably worse than those obtained with SiO_2_/Na_2_O molar ratio = 1.17, with the strengths decreasing by more than 90%. This behaviour suggests that the amount of dissolved silica is not the definitive parameter for yielding good strength development. The remaining alkalinity after thermochemical treatment plays a crucial role not only in the dissolution of the precursor, but also in the formation of the cementing N-A-S-H gel. Despite the low NaOH consumption, the final RHA-based activator properties were not appropriate for developing good strength mortars.

## 4. Conclusions

This study considers the use of an RHA-based activator in geopolymers, investigating dissolved silica by thermochemical treatments and reactions with NaOH and the strength of mortars prepared with an FCC precursor. The following main conclusions are drawn:

The filtering/gravimetric method for measuring dissolved silica after the thermochemical treatment is a good tool, even though a gelation process sometimes interferes with dissolved silica calculations, especially for ground RHA, for which a smaller particle size favours gel formation.

The four methods tested for dissolving silica from RHA (refluxing (R), high pressure and temperature (HPT), thermal bath at 65 °C (65C), and room temperature (RT)) are excellent procedures for preparing activators. Depending on the energy cost, the selected treatment time and the facilities available for preparing the activator on an industrial scale, one of these procedures could be selected.

The methods involving a reaction temperature over or equal to 65 °C resulted in silica dissolution percentages of about 80%. In RT, a percentage close to 70% was achieved. This behaviour means that a higher reaction temperature is needed to dissolve crystalline silica. 

The mechanical properties of the mortars prepared with RHA-based alternative activators were related to dissolved silica, and the gelation process did not affect the reactivity of the activator. 

Dissolved silica is not only the essential parameter to obtain a good activator; the remaining alkalinity also plays a crucial role in the mechanical development of geopolymeric mortars. Lowering the NaOH proportion in the activator did not affect silica dissolution, but diminished the solubilisation potential of the precursor.

## Figures and Tables

**Figure 1 materials-16-04667-f001:**
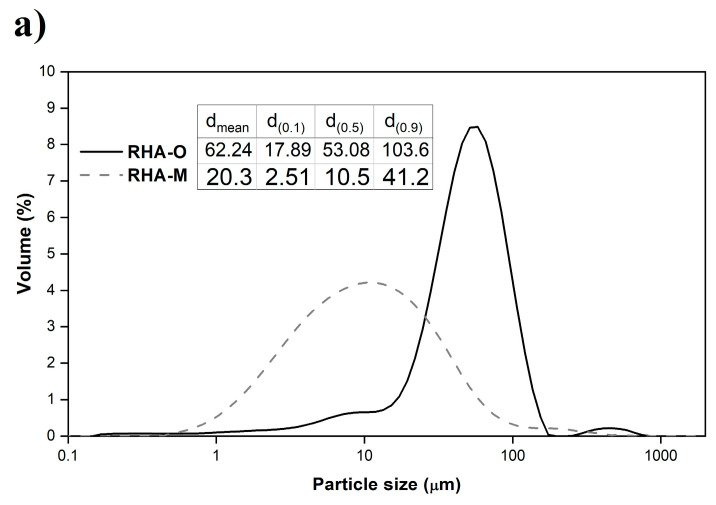
Characterisation of RHA: (**a**) granulometric distribution curves for RHA-O and RHA-M; (**b**) X-ray diffractogram of RHA-M.

**Figure 2 materials-16-04667-f002:**
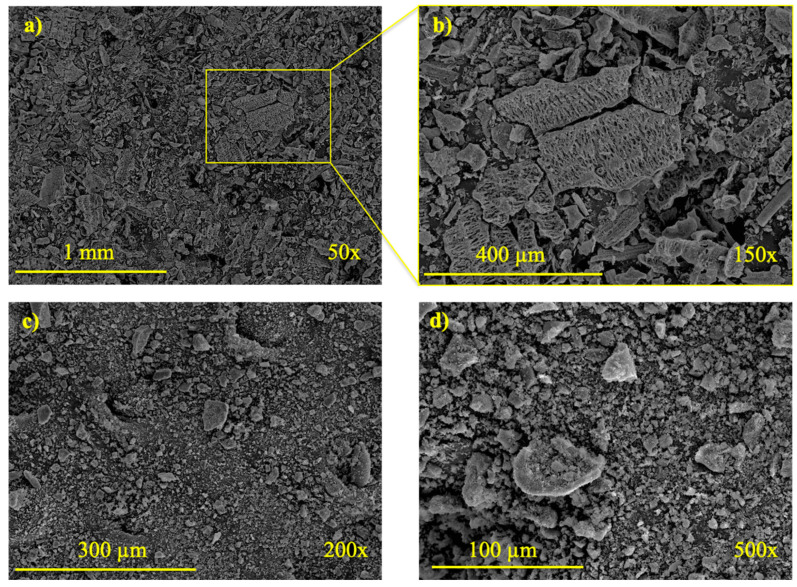
FESEM micrographs of used rice husk ashes: (**a**,**b**) RHA-O; and (**c**,**d**) RHA-M.

**Figure 3 materials-16-04667-f003:**
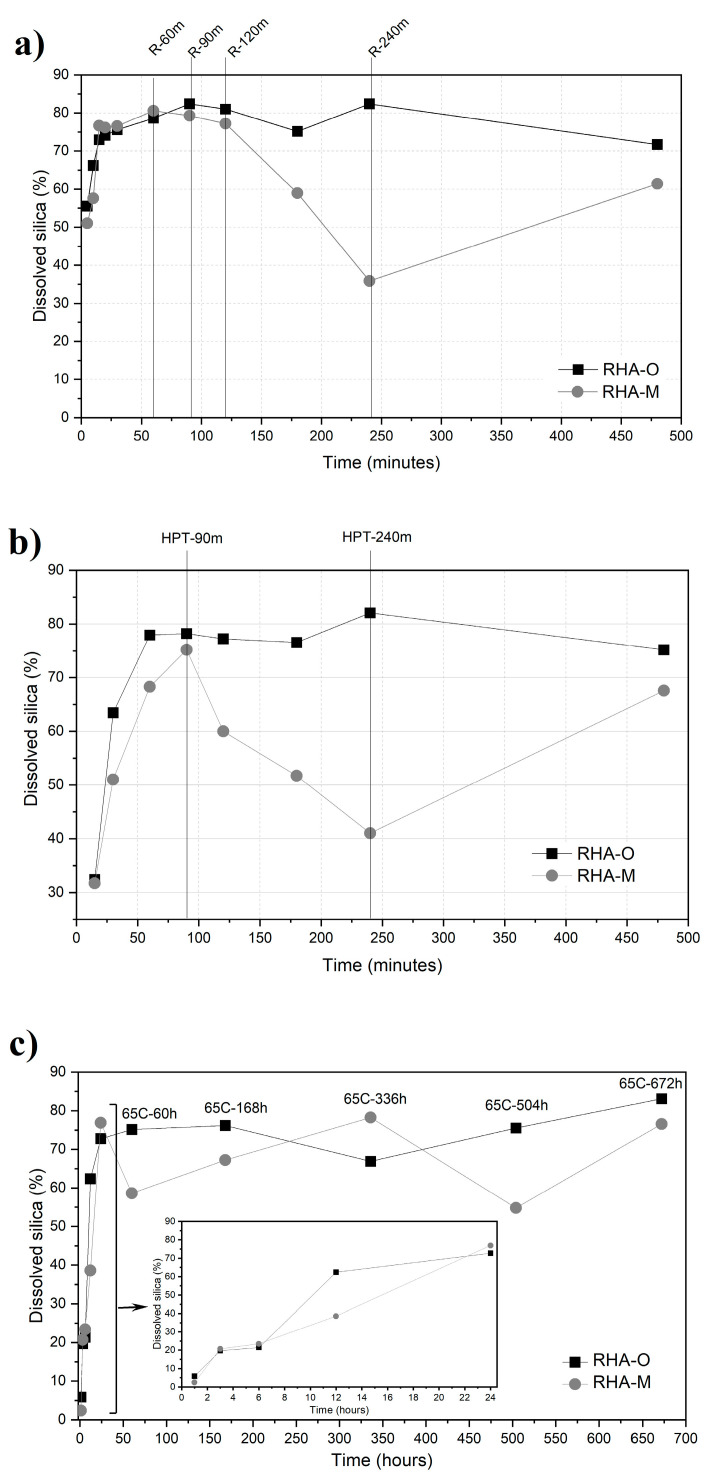
Graphical results of the dissolved silica obtained with each thermochemical treatment: (**a**) reflux; (**b**) high pressure and temperature; (**c**) thermal bath at 65 °C; and (**d**) room temperature.

**Figure 4 materials-16-04667-f004:**
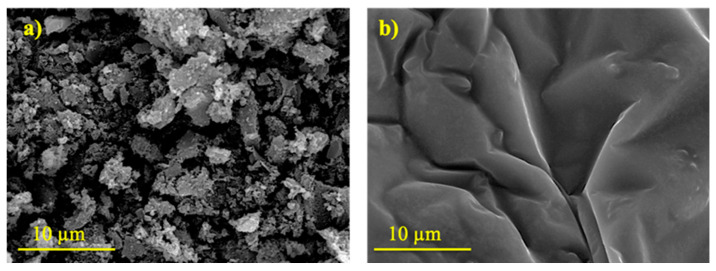
FESEM micrographs (×3500 magnification) of the solid residue obtained by R-240m for: (**a**) RHA-O; and (**b**) RHA-M.

**Figure 5 materials-16-04667-f005:**
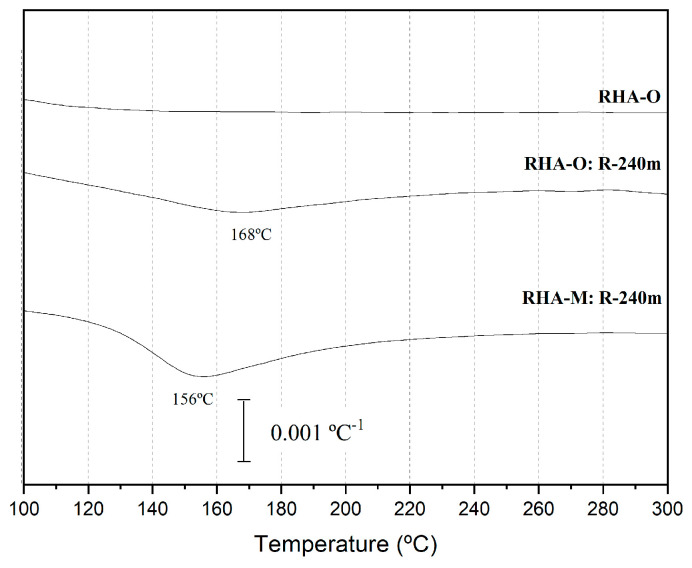
DTG curves of the retained solid after R-240m of RHA-O and RHA-M (the unprocessed RHA-O was used for comparisons).

**Figure 6 materials-16-04667-f006:**
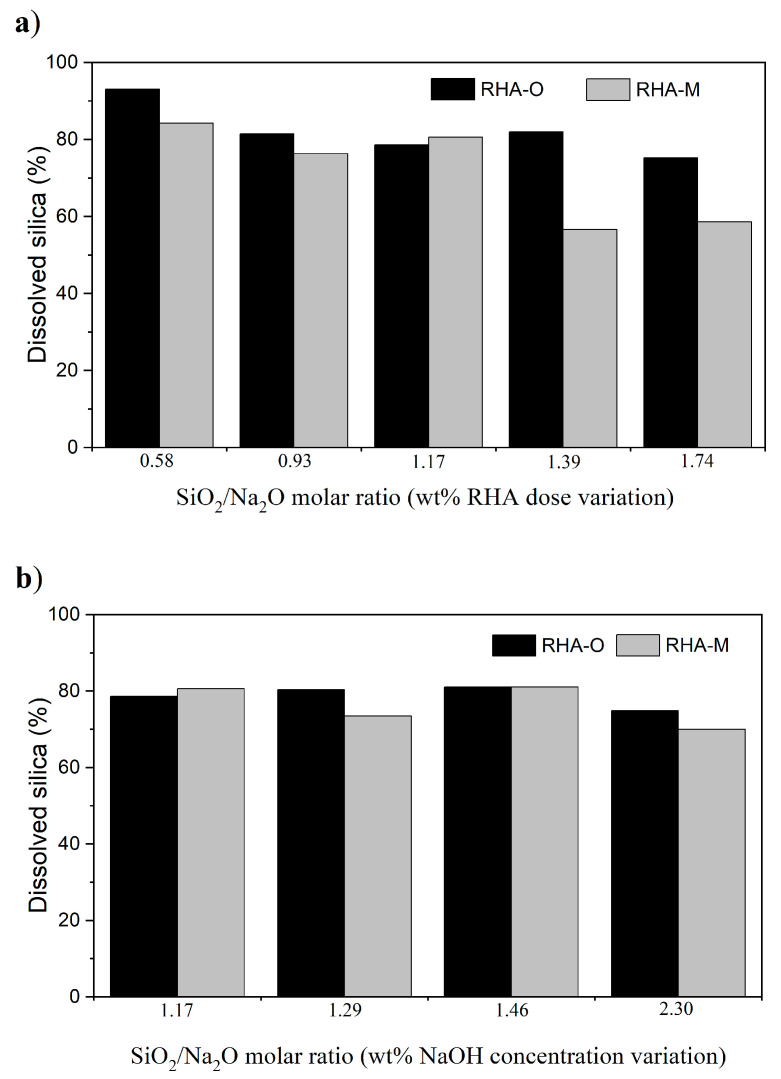
Study of the SiO_2_/Na_2_O molar ratio on the dissolved silica: (**a**) RHA variation; (**b**) NaOH variation.

**Figure 7 materials-16-04667-f007:**
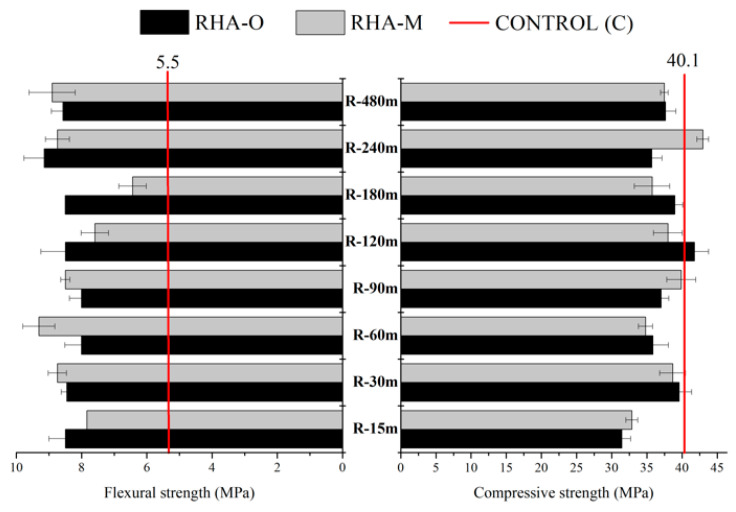
Mechanical behaviour after 24 h of curing at 65 °C for the mortars with the alternative RHA-based activator prepared by means of the reflux (R) treatment.

**Figure 8 materials-16-04667-f008:**
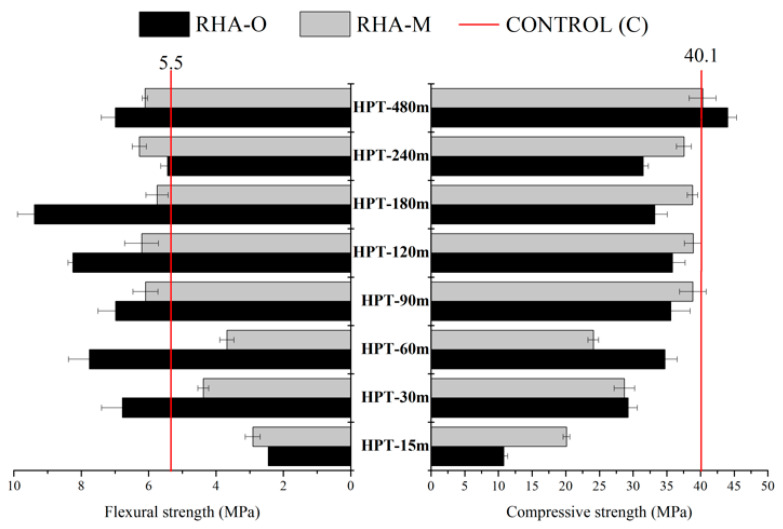
Mechanical behaviour after 24 h of curing at 65 °C for the mortars with an alternative RHA-based activator prepared by the high pressure and temperature (HPT) treatment.

**Figure 9 materials-16-04667-f009:**
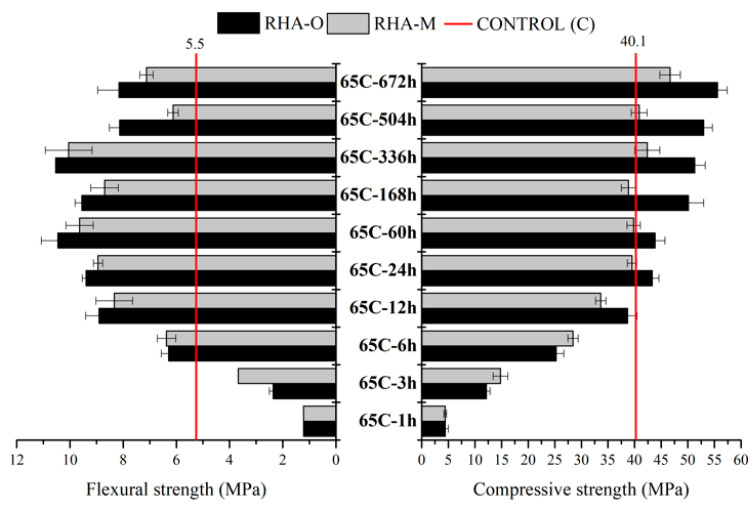
Mechanical behaviour after 24 h of curing at 65 °C for the mortars with an alternative RHA-based activator prepared by the thermal bath 65 °C (65C) treatment.

**Figure 10 materials-16-04667-f010:**
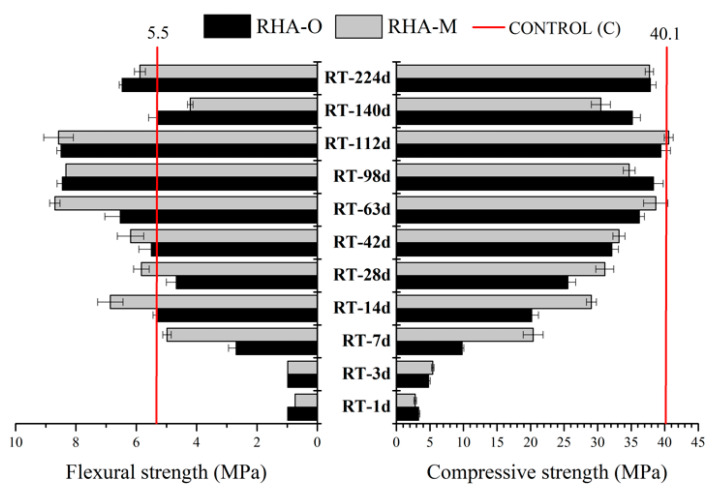
Mechanical behaviour after 24 h of curing at 65 °C for the mortars with an alternative RHA-based activator prepared by room temperature (RT) treatment.

**Figure 11 materials-16-04667-f011:**
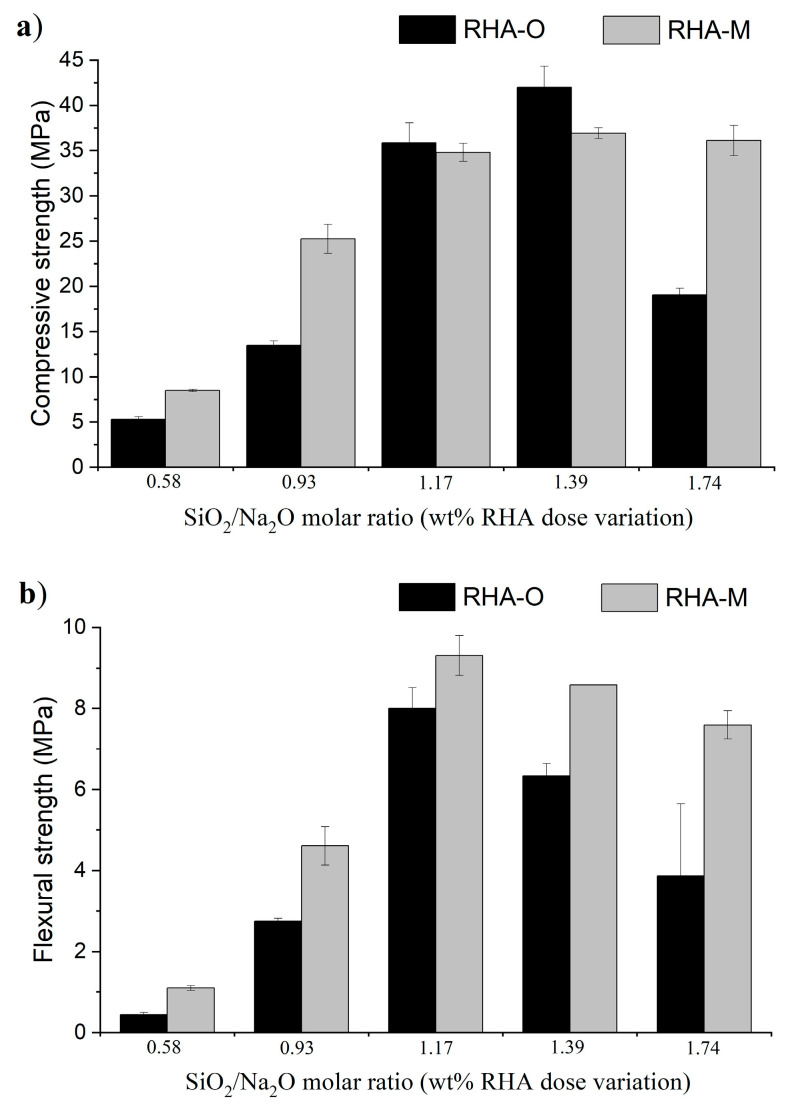
Study of the influence of the SiO_2_/Na_2_O molar ratio on the mechanical behaviour of mortars when varying the RHA dose in treatment R-60m: (**a**) compressive strength; and (**b**) flexural strength.

**Figure 12 materials-16-04667-f012:**
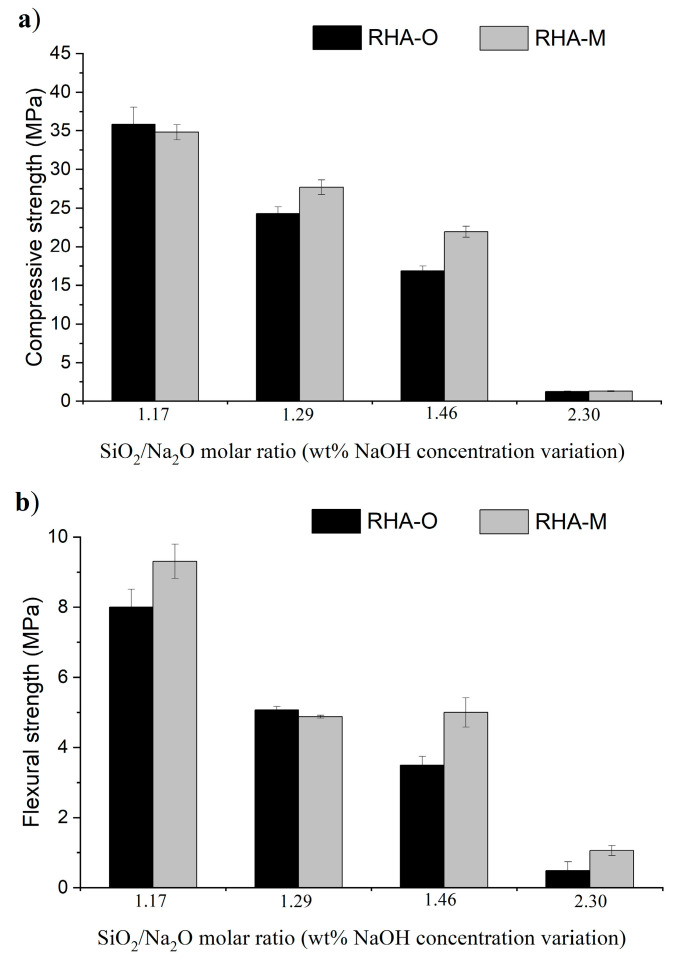
Study of the influence of the SiO_2_/Na_2_O molar ratio on the mechanical behaviour of mortars when varying the NaOH concentration in treatment R-60m: (**a**) compressive strength; and (**b**) flexural strength.

**Table 1 materials-16-04667-t001:** Chemical composition of FCC and RHA (wt%).

	SiO_2_	Al_2_O_3_	Fe_2_O_3_	CaO	MgO	SO_3_	K_2_O	Na_2_O	P_2_O_5_	Cl^−^	TiO_2_	LOI
FCC	47.46	49.26	0.6	0.11	0.17	0.02	0.02	0.31	0.01	-	1.22	0.51
RHA	85.58	0.25	0.21	1.83	0.5	0.26	3.39	-	0.67	0.32	-	6.99

**Table 2 materials-16-04667-t002:** Thermochemical treatments summary.

	Treatment	Acronym	Key	Treatment Time (X)	Units for Time
Dissolved RHA	Reflux	R	R-Xm	5, 10, 15, 20, 30, 60, 90, 120, 180, 240 and 480	Minutes
	High pressure and temperature	HPT	HPT-Xm	15, 30, 60, 90, 120, 180, 240 and 480	Minutes
	Thermal bath at 65 °C	65C	65C-Xh	1, 3, 6, 12, 24, 60, 168, 336, 504 and 672	Hours
	Room temperature	RT	RT-Xd	1, 3, 7, 10, 14, 21, 28, 35, 42, 56, 63, 98, 112, 140 and 224	Days
Mortar Manufacture	Reflux	R	R-Xm	15, 30, 60, 90, 120, 180 240 and 480	Minutes
	High pressure and temperature	HPT	HPT-Xm	15, 30, 60, 90, 120, 180, 240 and 480	Minutes
	Thermal bath at 65 °C	65C	65C-Xh	1, 3, 6, 12, 24, 60, 168, 336, 504 and 672	Hours
	Room temperature	RT	RT-Xd	1, 3, 7, 14 and 28	Days

**Table 3 materials-16-04667-t003:** Study of the SiO_2_/Na_2_O molar ratios: variation in RHA and variation in the amount of NaOH in the suspension prepared by treatment R-60m.

Variation	SiO_2_/Na_2_O Molar Ratio				
RHA	0.58	0.93	1.17	1.39	1.74
NaOH	1.17	1.29	1.46	2.30	

**Table 4 materials-16-04667-t004:** Dosage of control and RHA mortars.

Mortar	FCC (g)	RHA (g)	NaOH (g)	Na_2_SiO_3_ (g)	H_2_O (g)	Sand (g)
Control	450.0	_	54.8	253.1	108.0	1350.0
RHA-O or M	450.0	78.8	81.0	_	270.0	1350.0

**Table 5 materials-16-04667-t005:** Dosage for preparing the different activators used in the experimental program related to mortar assessment in the study on the influence of SiO_2_/Na_2_O molar ratio.

	SiO_2_/Na_2_O Molar Ratios	RHA (g)	NaOH (g)
Variation RHA	0.58	39.4	81.0
	0.93	63.0	81.0
	1.17	78.8	81.0
	1.39	94.5	81.0
	1.74	118.1	81.0
Variation NaOH	1.17	78.8	81.0
	1.29	78.8	72.9
	1.46	78.8	64.8
	2.30	78.8	40.5

## Data Availability

Not applicable.

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
