# Peer review of "Evaluation of Thermochemical Treatments for Rice Husk Ash Valorisation as a Source of Silica in Preparing Geopolymers"

_materials, 2023, doi:10.3390/ma16134667_

Round 1
Reviewer 1 Report
This is a very nicely written manuscript that may be published in the future. However, I propose to reject this paper because I do not like the evaluation of only mechanical properties after one day. In certain circumstances, we can consider the end of significant reactions in heat curing after only one day, but this is only in some cases and only for some applications. However, this article is not generally focused on this set of special applications. Especially with geopolymers, there is a need to study longer-term behavior and interaction with the real environment. The resulting material may have excellent properties after one day, but after a week in a real environment the properties may be poor and therefore it is insufficient to evaluate the quality of the product only after measuring the mechanical properties after 24 hours. For the publication of this manuscript, it would be necessary to include more information about the behavior of the resulting material over a longer period of time or really explain precisely why the properties are significant for this geopolymer only after 24 hours.
I have no essential comments to english language.
Author Response
Please, see attached file.
Thank you

Reviewer 2 Report
This work explores the use of RHA as an alternative sodium silicate reagent for geopolymer products. It employs four thermochemical treatments for silica dissolution from rice husk ash (RHA) and compares the silicon dissolving rate and the strength of geopolymer products made with RHA activatior.It is significant to find an alternative activator to replace sodium silicate as the alkali agent, as it is the main environmentally unfriendly component in this green material.The amount of data in the paper is not large, but the summary and analysis of the data are systematic.The experimental work was well organized, and the results were well discussed and interpreted.There are only a few minor parts that are suggested to be changed, as listed below:
Page 5 Line 245: Table 2 is quite far from this paragraph that provides an illustration for it. It is suggested to place the data and the illustration together.
Page 5 Line 262: It is necessary to use a table to compare the mix proportions of the mortars. Currently, it is not very clear whether the alkali activator is the only difference among these tested mortars.
Page 5 Line 262: It is suggested to show the photos of source materials, casting and finished samples here.
Page 12 Line 417: Is there any specific reason for choosing to test the flexural and compressive strength here? Usually, tensile strength is also a crucial mechanical characteristic for mortars. Additionally, some research indicates that the alkali activator has a significant influence on the tensile strength.
Finally, it is recommended to add microstructure images of the mortar reaction process using different activators to analyze the influence of RHA on geopolymerization.
no further comments.
Author Response
Please, see attached file.
Thank you

Reviewer 3 Report
Detailed comments are included in the attachment. Please revise accordingly to the suggestion.

Author Response
Please, see attached file.
Thank you

Reviewer 4 Report
Dear authors,
In this paper, the authors present a very interesting research related to the use of rice husk ash as an activator in the mortar recipe, concluding that it has a minimal influence on its mechanical properties.
As weak elements
The abstract can be extended.
In Figure 1, the legends, abscissas and ordinates are not clear.
For Figures 8-12, increase the font for the abscissa.
As a notable elements
It is a complete paper that contains experimental tests and ultramodern microstructural analyzes that can highlight with great precision the complex intercrystalline bonds between the different constituents, which gives the researcher a powerful tool for analysis and interpretation. Even if the results were not as expected, the value of the reduction is indisputably necessary and useful in the context of reducing pollution and producing new materials with a lower carbon footprint.
I can suggest to the authors in the future the use of design of experiment for a much more complex parametric analysis but with modern means of analysis and optimization of the costs of these experimental tests.
Author Response
Please, see attached file.
Thank you

Reviewer 5 Report
- Line 176. In Figure 1(b) it is recommended to add an X-ray diffractogram for RHA-O;
- Line 276. In Table 3 it is not clear what the numbers mean: 0.58, 0.93, etc. for RHA and 1.17, 1.29 etc. for NaOH;
- In section 2. "Materials and Methods", a table with the component composition for all experimental geopolymer mixes should be added;
- Lines 306–307. Behavior in the refluxed treatment, which ranged between 120 and 240 minutes (reduction in silica solubility) is not clear and not well explained in the text;
- Line 415. Figure 7. Based on the reported flexural and compressive strengths, the correlation between reaction time of the activator and its reactivity under the conditions of (R) treatment is not clear.
- Line 432–433. Please, explain, why for the thermochemical treatment, behavior of RHA-O is better vs. RHA-M?
Moderate editing of English language required
Author Response
Please, see attached file.
Thank you
